# Bap-Independent Biofilm Formation in *Staphylococcus xylosus*

**DOI:** 10.3390/microorganisms9122610

**Published:** 2021-12-17

**Authors:** Carolin J. Schiffer, Miriam Abele, Matthias A. Ehrmann, Rudi F. Vogel

**Affiliations:** 1Lehrstuhl für Technische Mikrobiologie, Technische Universität München, 85354 Freising, Germany; matthias.ehrmann@tum.de (M.A.E.); rudi.vogel@tum.de (R.F.V.); 2Bayerisches Zentrum für Biomolekulare Massenspektrometrie (BayBioMS), 85354 Freising, Germany; m.abele@tum.de

**Keywords:** *Staphylococcus xylosus*, knockout, Bap, biofilm, aggregation

## Abstract

The biofilm associated protein (Bap) is recognised as the essential component for biofilm formation in *Staphylococcus aureus* V329 and has been predicted as important for other species as well. Although Bap orthologs are also present in most *S. xylosus* strains, their contribution to biofilm formation has not yet been demonstrated. In this study, different experimental approaches were used to elucidate the effect of Bap on biofilm formation in *S. xylosus* and the motif structure of two biofilm-forming *S. xylosus* strains TMW 2.1023 and TMW 2.1523 was compared to Bap of *S. aureus* V329. We found that despite an identical structural arrangement into four regions, Bap from *S. xylosus* differs in key factors to Bap of *S. aureus*, i.e., isoelectric point of aggregation prone Region B, protein homology and type of repeats. Disruption of *bap* had no effect on aggregation behavior of selected *S. xylosus* strains and biofilm formation was unaffected (TMW 2.1023) or at best slightly reduced under neutral conditions (TMW 2.1523). Further, we could not observe any typical characteristics of a *S. aureus* Bap-positive phenotype such as functional impairment by calcium addition and rough colony morphology on congo red agar (CRA). A dominating role of Bap in cell aggregation and biofilm formation as reported mainly for *S. aureus* V329 was not observed. In contrast, this work demonstrates that functions of *S. aureus* Bap cannot easily be extrapolated to *S. xylosus* Bap, which appears as non-essential for biofilm formation in this species. We therefore suggest that biofilm formation in *S. xylosus* follows different and multifactorial mechanisms.

## 1. Introduction

*Staphylococcus (S.) xylosus* is a Gram-positive, coagulase-negative commensal of mammal skin with a biotechnological relevance, as it is commonly used as a starter organism for raw sausage fermentation [1]. For persistence in such environments, surface colonization and the formation of biofilms are anticipated as important traits. Biofilm formation is a common property of many bacteria and describes a state where cells are embedded in an extracellular matrix, mainly consisting of exopolysaccharides, proteins and extracellular DNA [2]. Biofilm matrix composition can vary and is dependent on species-specific mechanisms as well as environmental conditions [3]. If biofilms are of a mainly proteinaceous nature, surface proteins play a prominent role in primary adhesion and biofilm maturation as they can either interact with surface structures on adjacent cells or form amyloid structures that promote cellular aggregation [4]. Known surface proteins influencing proteinaceous biofilm matrix assembly include fibronectin binding proteins (FnBPs), *Staphylococcus aureus* surface protein G (SasG) and the biofilm associated protein (Bap) [5,6]. Bap is a high molecular weight surface protein, which was first described by Cucarella et al. [7]. Bap and its homologues have been shown to mediate multicellular aggregation as well as biofilm formation in several organisms, such as staphylococci and enterococci [8,9]. In *S. aureus*, V329 Bap is extensively studied, and it has been demonstrated that its functionality is based on self-assembly of *N*-terminal peptides of the protein into amyloid fibers under acidic conditions [9]. Furthermore, Bap-mediated biofilm formation is influenced by calcium ions in the environment, as they prevent amyloid assembly of Bap-derived peptides and subsequent intercellular aggregation [10]. Other than in the well-studied *S. aureus* V329, Bap function has only been molecularly characterized by the construction of knockout mutants in *S. epidermidis* C533 [11]. Studies addressing *bap* of biofilm-positive *S. xylosus* strains have been focusing solely on presence/absence of the gene analysed by PCR/hybridization approaches so far [11,12]. Thus, the actual function of the protein in *S. xylosus* has not been verified experimentally yet.

In a previous study, we demonstrated a strain-specific behavior in biofilm formation among different *S. xylosus* strains depending on environmental conditions as well as their individual genomic settings with respect to additional biofilm-related genes [13]. The fact that a strain in which *bap* is naturally defective displayed a biofilm negative phenotype led us to suggest an essential role of Bap in biofilm formation of *S. xylosus.* In order to test this hypothesis and to ensure that any impact of other genomic determinants is excluded, we generated isogenic *bap* knockout mutants of biofilm positive strains. Furthermore, we investigated whether *S. xylosus* and its *bap* mutants show typical phenotypic characteristics and differences, as they have been reported in Bap-positive *S. aureus* strains and its respective *bap* mutants.

## 2. Materials and Methods

### 2.1. Bacterial Strains and Culture Conditions

*S. xylosus* TMW 2.1023 and TMW 2.1523 are both biofilm-positive strains, isolated from raw fermented sausages [13]. *Escherichia (E.) coli* DC10B is a cytosine methyltransferase-negative derivate, often used in transformation experiments to evade type IV restriction modification systems [14].

*E. coli* DC10B was cultured in Lysogeny Broth (LB, tryptone 10 g/L, yeast extract 5 g/L, NaCl 5 g/L) 37 °C. Staphylococcal strains were cultured at 28 or 37 °C in Trypticase soy broth (TSB, casein peptone 15 g/L, soy peptone 15 g/L, yeast extract 3 g/L) supplemented with either no glucose (TSB_N_) or 1% glucose (TSB^+^), brain heart infusion (BHI) broth, or basic medium (BM, 1% peptone, 0.5% yeast extract, 0.5% NaCl, 0.1% Glucose, 0.1% K_2_HPO_4_). For transformation experiments, 20 µg/mL (*E. coli*) or 10 µg/mL (*S. xylosus*) of chloramphenicol (CarlRoth, Karlsruhe, Germany) were added when necessary.

### 2.2. DNA Manipulations and Bacterial Transformation: Mutagenesis of the Chromosomal Bap Gene by Allelic Exchange

For inactivation of *bap* in *S. xylosus*, regions up—and downstream of the sequence to be deleted were amplified using primers bap1F (5′-ACTCACTATAGGGCGAATTGGAGCTGTTATCAGCAGCTGCTAAG-3′), bap2R (5′-GTATATTGCGACACAATGTAAAGTATATCAG-3′), bap3F (5′-CTTTACATTGTGTCGCAATATACAGCTAG-3′) and bap4R (5′-GCTTGATATCGAATTCCTGCAGCATCTATAACTTTAGCTG-3′). To be able to use the same primer set for both *S. xylosus* strains (TMW 2.1023, TMW 2.1523), primers were chosen to map on conserved regions of the gene. PCR products were purified using a Monarch PCR and DNA cleanup kit (New England Biolabs (NEB), Ipswich, United Stated). Shuttle vector pIMAY*, which was kindly provided by A. Gründling (Molecular Microbiology, Imperial College London, UK) was digested with restriction enzymes *SacI* and *PstI* and PCR fragments were ligated into the vector using Gibson Assembly (NEB). The construct was transformed into *E. coli* DC10B by electroporation and successful transformants were selected on chloramphenicol plates. Sequencing, to verify correct assembly of the vector, followed. Plasmid was isolated using the Monarch plasmid DNA miniprep kit (NEB) and transformed into electrocompetent *S. xylosus* cells as described by Monk et al. [15]. Briefly, *S. xylosus* was cultured overnight in BHI and diluted to an OD_600_ of 0.5 in BM the next morning. Cultures were then incubated at 37 °C and 200 rpm for another 40 min, harvested, washed twice with ice-cold water and another two times with 10% glycerol in decelerating volumes (1/10, 1/25). Finally, cells were resuspended in 1/200 volume 10% glycerol + 500 mM sucrose and directly subjected to electroporation. At least 1 µg of plasmid was transformed (0.2 cm, 2.5 kV) and cells were immediately suspended in 1 mL BHI + 200 mM sucrose. After one hour at 28 °C, cells were spread on BHI 20CM and incubated at 28 °C for two days. Colonies were picked and allelic replacement was performed as described by Schuster et al. [16]. Lastly, successful allele replacement of the chromosomal *bap* sequence as well as plasmid loss were verified by colony PCR using primers bap1F and bap4R.

### 2.3. Colony Morphology on CRA

Colony morphology of wildtype and mutant strains on congo red agar (CRA, 10 g/L glucose, 0.8 g/L congo red) was assessed as previously described [17]. As congo red interacts with proteins and proteinaceous structures, strains with rough colony margins were considered as Bap positive, whereas Bap negative strains usually retain smooth colony margins.

### 2.4. Biofilm Formation Assays

Biofilm formation was quantified in 96-well plates as described by Schiffer et al. [13]. Strains were cultured in different media (TSB_N_, TSB^+^ (pH 7.2), Lac^+^ (TSB^+^ acidified with lactic acid to pH 6.0) and on different supports (hydrophobic (Sarstedt, Nürmbrecht, Germany), hydrophilic (Nunclon^TM^ delta, Thermo scientific, Waltham, MA, USA)). The media chosen, represent the three media found in a previous study to influence biofilm formation of the selected strains the most [13]. Lactic acid was used instead of an inorganic acid, as it is a prevalent acid found in one of the habitats of the species *S. xylosus* (raw fermented sausages). To determine whether calcium influences biofilm formation, CaCl_2_ was added to the wells to a final concentration of 20 mM when indicated.

### 2.5. Bacterial Aggregation Assay

Bacterial aggregation was determined by growing cell suspensions (OD_600_ at t_0_: 0.1) in test tubes for 24 h at 37 °C, 200 rpm in either TSB_N_ or TSB^+^. Aggregation behavior was evaluated macroscopically. To investigate whether calcium has an impact on cellular aggregation behavior, CaCl_2_ was added to a concentration of 20 mM when indicated.

### 2.6. Growth and pH Dynamics

pH changes of *S. xylosus* strains in TSB^+^ were monitored using the icinac system (AMS Systea, Rome, Italy). Growth curves were recorded by determining the optical density over a period of 33 h in a Microplate Reader (Spectrostar^Nano^, BMG Labtech, Ortenburg, Germany).

### 2.7. SDS Page of Protein Extracts

Bacteria were grown in TSB_N_ until early stationary phase (37 °C, 200 rpm, 12 h). Then, 5 mL of cell culture were harvested (4 °C, 5000× *g*), washed twice with ice-cold phosphate-buffered saline (PBS) and resuspended in 150 µL PBS + 30% [wt/vol] raffinose (SigmaAldrich, St. Louis, MI, USA). Afterwards, 7 µL lysostaphin (1 mg/mL, SigmaAldrich, St. Louis, MI, USA) and 3 µL of DNAseI (1 mg/mL, SigmaAldrich, St. Louis, MI, USA) were added and after incubation at 37 °C for 2 h, protoplasts were sedimented at 8000× *g* for 30 min with slow deceleration. Supernatants were stored at −20 °C until subjection to SDS-PAGE analysis (10% resolving, 4% stacking gel).

### 2.8. Full Proteome Analysis 

For full proteome analysis of *S. xylosus* cells, 0.1% of overnight cultures were diluted in fresh Lac^+^ and incubated in Erlenmeyer flasks under agitation (planktonic, 5 mL) or statically in Nunclon^TM^ delta surface (Thermo scientific, Waltham, MA, USA) tissue culture plates (sessile, 2 mL) for 24 h at 37 °C. Cell lysis and in-solution digest were performed with minor changes according to the SPEED protocol [18]. Shortly, 2 mL of planktonic cells were harvested, washed twice with ice-cold PBS and resuspended in 100 µL of absolute Trifluoroacetic acid (TFA). Sessile cells were also washed twice with ice-cold PBS to remove any non-adherent cells as well as to remove as many media proteins as possible, and dissolvement of adherent cells was performed by carefully resuspending the biofilm in 100 µL (TFA). All TFA cell suspensions were neutralized to pH 8.1–8.3 by adding nine volumes of Tris (2 M, pH not adjusted). Incubation in a thermomixer for 5 min at 55 °C and 450 rpm (ThermoMixerC, Eppendorf, Hamburg, Germany) as well as short centrifugation followed. Protein concentrations were determined using Bradford assay (B6916, SigmaAldrich, St. Louis, MI, USA) according to manufacturer’s instructions. Then, 15 µg of total protein amount were reduced, alkylated (10 mM Tris-(2-carboxyethyl)-phosphin, 40 mM 2-Chloroacetamide; 5 min, 95 °C) and afterwards diluted with water (1:1). Trypsin digest was performed in an enzyme to protein ratio of 1:50 overnight at 30 °C with mild agitation (400 rpm) and then stopped with 3% Formic acid (FA). Three discs of Empore C18 (3 M, Saint Paul, Minnesota, United States) material were packed in 200 µL pipette tips. The resulting desalting columns were conditioned (100% acetonitrile, can) and equilibrated (40% ACN/0.1% FA followed by 2% ACN/0.1% FA). Peptides were loaded, washed (2% ACN/0.1% FA) and eluted (40% ACN/0.1% FA). For the determination of expression levels of Bap in sessile versus planktonic cultures, around 250 ng peptides of three biological replicates were subjected to an Ultimate 3000 RSLCnano system coupled to a Q-Exactive HF-X mass spectrometer (Thermo Fisher Scientific, Waltham, MA, USA). All acquisition parameters were the same as described by Kolbeck et al. [19]. Proteomic analysis of the mutant strains was done in single measurements. Around 250 ng of peptides were subjected to an Ultimate 3000 RSLCnano system coupled to a Fusion Lumos Tribrid mass spectrometer (Thermo Fisher Scientific, Waltham, MA, USA). All parameters were set as described by Bechtner et al. [20].

Since TMW 2.1523 is a well biofilm former in TSB_N_ [13], it was chosen that planktonic as well as sessile data was sampled for this particular strain not just in Lac^+^ but also in TSB_N_.

### 2.9. Bioinformatic and Statistical Analysis

Bioinformatic analysis and comparative genomics were performed using CLC Main Workbench 8 software (CLC bio, Aarhus, Denmark). Isoelectric point (pI) and molecular weight (MW) were computed using the respective tool from the Expasy server (available under: https://web.expasy.org/compute_pi/, accessed on 29 November 2021), InterPro (86.0) was used to predict signal peptide, transmembrane regions and cell wall anchor and ProScan (screens against PROSITE database) was used to scan for EF-hand motifs (cut off was set to 80% protein identity). Amyloidogenic regions in protein sequences were analyzed by using the amyloid finder tools FoldAmyloid [21], Aggrescan [22], Waltz-DB 2.0 [23] and Tango [24].

Peptide identification and quantification were performed using MaxQuant (v1.6.3.4) with Andromeda^97^ [25,26]. MS2 spectra were searched against the NCBI proteome databaste of *S. xylosus* 2.1023 and 2.1523, respectively; common contaminants were included (built-in option in MaxQuant). Trypsin/P was specified as proteolytic enzyme. Precursor tolerance was set to 4.5 ppm and fragment ion tolerance to 20 ppm. Results were adjusted to 1% false discovery rate (FDR) on peptide spectrum match level and protein level employing a target-decoy approach using reversed protein sequences. Minimal peptide length was defined as 7 amino acids; the “match-between-run” function disabled. Carbamidomethylated cysteine was set as fixed and oxidation of methionine and *N*-terminal protein acetylation as variable modifications. Perseus version 1.6.15.0 [27] and LFQ-Analyst [28] were used for data analysis. Missing label-free quantitation (LFQ) values were imputed from normal distribution. Significant differences in intensities between the conditions chosen, were calculated by student’s t-test with α set to 0.05 FDR correction (Benjamini Hochberg method) was applied to correct *p*-values. Data are, if not otherwise indicated, presented as means +/− standard errors of the means. Student’s *t*-tests were performed using Perseus Version 1.6.15.0.

## 3. Results

### 3.1. Protein Motif Structural Organization of S. xylosus Bap

To allow conclusions on structure-function correlations, the protein structures of *S. xylosus* and *S. aureus* Bap were compared. To get an idea of the conservation of the protein along the full sequence length, each region was aligned separately. Hereby, we sticked to the four major regions of the protein which were defined previously [7,13]. Bap of *S. xylosus* TMW 2.1023 and TMW 2.1523 share high sequence similarities, with 99.4% (Region A), 99.4% (Region B), 79.5% (Region C) and 98.4% (Region D) protein identity, respectively. Strain specific differences of *S. xylosus* Bap mainly rely on the number of C-Repeats, which have previously been shown to have no impact on the biofilm function of Bap [30]. Therefore, both *S. xylosus* sequences are referred to as Bap_XYL_ in this paragraph and compared to the sequence of the phenotypically well-characterized, Bap-positive strain *S. aureus* V329 (Bap_AUR_).

Bap_XYL_ as well as Bap_AUR_ are both high molecular weight proteins fulfilling typical criteria of surface proteins such as a 44 amino acid (aa) long *N*-terminal signal sequence (YSIRK motif) as well as a C-terminal hydrophobic transmembrane segment and the LPxTG cell wall anchor motif. However, deeper sequence comparison of Bap of these two species revealed some notable differences (Table 1).

While the predicted pI of the full protein sequences is almost identical (~3.9), it is noticeably lower in Bap_XYL_ when the aggregation prone Region B (BapB_AUR_: 4.6, BapB_XYL_: 4.4) is considered only. Further, repeating sequence patterns differ remarkably between the proteins of both species. While Bap_AUR_ carries two large repeating sequences in region A, BapA_XYL_ contains only two short tandem repeats. Similar applies for region D repeats. BapD_AUR_ repeats are rich in serine and aspartate, while BapD_XYL_ repeats are not just shorter but also lacking these amino acids and are rather rich in glycine. Another difference is the type of repeats in the C Region of the protein described for both species. For BapC_XYL_ repeats are predicted to be ig-like domain type 6 repeats, while for BapC_AUR_ repeats are characterized as ig-like domain type 3 repeats. It is also noteworthy that Bap_XYL_ carries almost twice as many predicted EF hand binding motifs than Bap_AUR_. However, EF hand motifs EF2 and EF3 displayed by Region B of the protein, which have been shown to have a regulating effect on the activity of the protein [10], are identical in both species.

Several online tools were employed to predict amyloidogenic regions of the three protein sequences. All proteins share a region with high amyloidogenic potential, roughly between amino acid 400 and 900 (Figure 1).

However, while this region seems to be the only one with amyloidogenic potential in *S. aureus* V329, results are less conclusive for Bap_XYL_, as, especially in TMW 2.1523, regions with amyloidogenic structural characteristics are also predicted in the C-terminal part of the protein. For Bap_AUR_, two peptides (I: _487_TVGNIISNAG_496_, II: _579_GIFSYS_584_) are characterized as displaying significant amyloidogenic potential [9]. *S. xylosus* harbors similar peptide sequences in its Bap sequence, however, not identical (I: _490_TVANILNNAG_499_, II: _582_GVFSYS_587_). To ensure that the sequences of our selected *S. xylosus* strains are representative for the species, we compared them to a range of other *S. xylosus bap* sequences as well (see Appendix A). In this context we also included the sequence of *S. epidermidis* C533 into the alignments, as Tormo et al. [11] have previously reported a biofilm negative phenotype for this strain upon deletion of *bap*. Sequence analysis confirmed that *S. aureus* and *S. epidermidis* Bap are closely related and share high sequence homologies in all parts of the protein. *S. xylosus bap* sequences on the other hand, share high homologies among themselves, but differ from *S. aureus* and *S. epidermidis bap* genes in some functionally important parts. Namely, they lack long amino acid repeats in region A, harbor slightly different amyloidprone peptide sequences in region B, carry different types of C-repeats and are rather composed of G-rich as to SD-repeats in region D. It is further worth noting that we found a high range of truncated Bap sequences among *S. xylosus* strains, and also in biofilm positive strains ([12], e.g., *S. xylosus* C2a, accession: WP_144404228.1). This substantiates our hypothesis that the protein is non-essential for biofilm formation by this species.

### 3.2. Mutagenesis of the Chromosomal Bap Gene

For mutagenesis of the chromosomal *bap* gene, 452 nt of the *bap* promotor region as well as the first 1427 nt of the *N*-terminal part of the protein in *S. xylosus* strains TMW 2.1023 and TMW 2.1523 were deleted. To confirm successful knockout of the protein, cell extracts were at first subjected to SDS-PAGE analysis. For TMW 2.1523 a weak band was visible at the expected position for Bap (224 kDa) in the wildtype strain (indicated by arrow) but not in the mutant strain (Appendix A). For TMW 2.1023 no band was detectable at the position of Bap (173 kDa) in neither wildtype nor mutant strain.

Since visibility of *S. xylosus* Bap on SDS gels was poor and inconclusive namely for strain TMW 2.1023, a high-sensitivity mass-spectrometric analysis of the full proteome was performed. Bap deletion was confirmed by comparing the peptide sequences mapping to Bap in wildtype in contrast to mutant samples. No peptide, except for one in TMW 2.1023 mutant, was found in the full proteome analysis (Appendix A). This result confirmed successful knockout of *bap*.

### 3.3. Growth Dynamics of Wildtype and Mutant Strains

To ensure that the transformation procedure led to no growth defects as well as to investigate whether Bap deletion has an impact on the growth behavior of *S. xylosus*, growth curves in different media were recorded. Figure 2 shows the growth dynamics for both strains in TSB_N_ (pH 7.2), TSB^+^ (pH 7.2), Lac^+^ (pH 6) and TSB^+^-HCl (pH 6). While wildtype and mutant strains behave very similar in TSB_N_ and TSB^+^, curves show higher variation when the growth medium is acidified to pH 6 by either the addition of lactic acid or HCl from the very beginning. However, in general one cannot ascribe any growth deficiencies to the *bap* mutant strains.

### 3.4. Biofilm Formation of Bap Wildtypes and Mutants

To determine the impact of Bap on biofilm formation of *S. xylosus*, we tested the biofilm forming capacities of two wildtype strains and their respective mutants in a 96-well plate assay. Tests were performed on hydrophilic and hydrophobic supports as well as in three different growth media that have previously been shown to promote biofilm formation to various extents in these strains [13]. The results of the biofilm assay are depicted in Figure 3. A significant reduction of biofilm formation on both supports was detectable with strain TMW 2.1523 in TSB_N_ only, thus in a medium where no sugar is present and therefore no pH decrease occurs upon growth. For strain TMW 2.1023, no differences in biofilm formation between wildtype and *bap* mutant strain were observed.

### 3.5. Colony Morphology of Bap Wildtype and Mutant Strains on Congo Red Agar

A typical characteristic of Bap positive *S. aureus* strains is their rough colony morphology on congo red agar as the dye interacts with proteinaceous, fibrillar structures [31]. Loss of Bap causes a transformation of their rough colony morphologies to a smooth type [7]. As shown in Figure 4, we could not observe any differences in colony morphology between wildtype and mutant strains on congo red agar in *S. xylosus*. Even after a couple of days of inoculation, no switches to Bap positive phenotypes (rough colonies) were observable and colony margins remained smooth.

### 3.6. pH Changes of S. xylosus during Growth in Glucose Supplemented Media

In *S. aureus* the formation of Bap-based amyloid structures starts with the entry of *S. aureus* cells into stationary phase when the pH drops under pH 5 [9]. The mechanism of Bap-based aggregation is very sensible to pH changes, best functioning when the environment reaches pH values close to the isoelectric point of the protein. However, pH measurements over time of *S. xylosus* strains in TSB^+^ show, that *S. xylosus* is not lowering the pH below 5.0 (see Appendix A). After 24 h of pH measurement, *S. xylosus* incubated in TSB^+^ yielded values between 5.1 ± 0.2 and 5.0 ± 0.1 for TMW 21023 and TMW 2.1523, respectively. The observed accelerated acidification of TMW 2.1023 compared to TMW 2.1523 is in concurrence with the observed, faster growth of this particular strain.

### 3.7. Calcium Does Not Impair Biofilm Formation of Bap Positive S. xylosus Wildtype Strains

Since calcium has been described as a negative effector on biofilm formation in *S. aureus* V329 and addition of calcium to the growth medium completely abolished Bap mediated biofilm formation [10], the role of Ca^2+^ on *S. xylosus* biofilm formation was investigated. Therefore, CaCl_2_ was added to a concentration of 20 mM to the wells and biofilm formation was quantified again. As shown in Figure 5 (the respective staining results are shown in Appendix A), no biofilm-eradicating effect of calcium was found on neither of the tested supports nor in any of the tested growth media. This result leads us to hypothesize that Bap is not a major factor of *S. xylosus* biofilm formation in these strains.

### 3.8. Formation of Cell Aggregates in Wildtype and Mutant Strains

Previous studies with *S. aureus* V329 have shown that Bap is engaged in intercellular interactions, promoting aggregate formation under acidic conditions [9]. We found that *S. xylosus* is prone to aggregation in an acidic environment, too. However, neither the absence of Bap nor addition of CaCl_2_ can impair cell aggregation as it has been described for *S. aureus* V329. As shown in Figure 6, all cultures showed heavy cell clumping with cells precipitating either at the bottom of the tube or building a top layer at the air-liquid interface when incubated in TSB^+^.

Generally, the formation of multicellular aggregates is much weaker when strains are incubated in TSB_N_, where pH remains in a neutral to basic range. Still, in neither of the media tested, any differences between wildtype and mutant strains were detectable. We further tested the aggregation and biofilm behavior of *S. xylosus* strains when incubated in medium acidified with 1 M HCl to pH 4.5. However, no growth was detectable and thus neither aggregation nor biofilm formation could be observed. Acidification to pH 4.5 of the cells, which had been grown at neutral pH, did not lead to a change to a clumping phenotype either. Cells remained in a turbid suspension, comparable to when they were grown under neutral conditions (TSB_N_).

### 3.9. Proteomic Analysis of Expression Levels of Bap under Planktonic versus Sessile Conditions

Since visibility of Bap on SDS page was poor, we decided to use a full proteome analysis to confirm that Bap is expressed and to monitor the expression of Bap under different conditions. Therefore, we compared the measured intensity values for Bap of both *S. xylosus* strains when planktonic growth in Lac^+^ occurred compared to sessile growth in the same medium. Hereby, detected Bap amounts did not change significantly (TMW 2.1023) or only to a minor degree (1.15 log_2_ fold, TMW 2.1523) between the two compared conditions; however, one should keep in mind that cells grown planktonically in Lac^+^ show heavy cell aggregation which is closely related to biofilm formation. On the other hand, when Bap amounts are compared in cells grown in TSB_N_, the protein shows a more than fivefold reduction in biofilm stages compared to planktonic growth (Table 2, Appendix A, note: TSB_N_ data only available for TMW 2.1523).

Further, Bap expression is significantly higher (+2.97) when TMW 2.1523 planktonic cells are grown in TSB_N_ compared to Lac^+^.

In order to obtain a better understanding of whether Bap of *S. xylosus* is a protein with a generally low abundance in the cell, Bap intensity values were not just compared between samples obtained from cells at different growth stages but also within samples. Plotting all protein intensities measured in one sample under one condition, reveals that Bap is not a highly abundant protein such as, for example, ribosomal proteins are. However, Bap expression is not remarkably low either (compare Appendix A).

## 4. Discussion

In this study different experimental approaches were used to evaluate the impact of Bap in *S. xylosus* on its anticipated general role in biofilm formation and multicellular behavior. Therefore two *S. xylosus* wildtype strains were chosen, that have previously been shown to display different biofilm positive phenotypes [13]. After genetic and phenotypic comparison of the two strains and their isogenic *bap* mutants we found that the role of Bap in *S. xylosus* is most likely non-essential for biofilm formation, apparently strain-specific, and predictively more diverse than anticipated. Upon deletion of *bap*, a significant reduction in biofilm formation was only observed with *S. xylosus* TMW 2.1523 under neutral pH environmental conditions. It did not affect its aggregation behavior and had no effect on any of the investigated traits of *S. xylosus* TMW 2.1023. This is in contrast to studies on *S. aureus* V329 [7] and *S. epidermidis* C533 [11], which showed that disruption of *bap* led to complete abolishment of biofilm formation. We therefore conclude that, firstly, *bap* orthologs do not necessarily have the same function in other staphylococcal species as the one previously determined for single strains of *S. aureus* and *S. epidermidis* and secondly, that biofilm formation and aggregation of *S. xylosus* involves different mechanisms. This view is supported by our previous study, which revealed that different *S. xylosus* strains encode a variable set of biofilm related genes in their genomes and display a biofilm phenotype that is dependent on environmental conditions in a strain-specific manner [13]. It is also supported by the observation that *S. xylosus* C2a has been described as a strong biofilm producer [12], even though we found that Bap is truncated by a premature stop codon in region B in this strain. Given this, it appears likely that either other factors dominate biofilm formation of the species, such as eDNA, which has been named as an important component of *S. xylosus* C2a biofilm matrix [32], or that Bap does not have a predominant role in *S. xylosus* biofilm formation, due to either low expression or functional differences of the protein.

Congo red is commonly used as amyloid dye and has been shown to interact with amyloidogenic structures formed by the *N*-terminal part of Bap [9,31] resulting in rough colony phenotypes of Bap positive strains on CRA. Loss of Bap has subsequently been shown to transform rough colonies to colonies with smooth margins [31]. Even though *S. xylosus* strains TMW 2.1023 and TMW 2.1523 are Bap positive, no colony morphology changes in wildtype versus mutant strains could be observed. Both strains and their respective mutants displayed smooth colony margins, suggesting that in contrary to Bap positive *S. aureus* strains, (Bap based-) amyloid formation seems to be different or at least under the conditions tested, not existent in *S. xylosus*. Another factor supporting a minor role of Bap in biofilm formation of *S. xylosus* is that calcium addition (20 mM) had neither an inhibiting effect on cell aggregation in culture tubes nor on biofilm formation. In *S. aureus* V329 adding as little as 6 mM calcium to the growth medium already abolishes any kind of aggregation and biofilm formation behavior [10]. If Bap binds calcium ions with low affinity, the molten globule conformation of the protein is stabilized in a way that no subsequent assembly to amyloid structures occurs. The effect of calcium binding and stabilization is attributed to EF domains 2 and 3 in the B region of the protein. However, even though Bap_XYL_ harbors the exact same EF hand motifs in its B region as Bap_AUR_, no aggregation/biofilm reducing effects were observed upon calcium addition. On the contrary, in some cases one could rather predict a trend towards an enhancing effect on biofilm formation especially on hydrophobic support. If and to which extent the increased number of EF hand motifs found in Bap_XYL_, compared to Bap_AUR_, is related to this observation remains speculative.

pH plays an important factor in Bap-mediated biofilm formation, as self-assembly of the protein into amyloid fibers occurs under acidic conditions only. When *S. aureus* V329 is incubated in medium containing glucose, pH drops below 5 as soon as stationary phase is entered, thereby approaching the pI of the aggregation prone B region of the protein, which facilitates amyloid structure formation [9]. *S. xylosus* differs in two points. First, the calculated pI of Region B of BapB_XYL_ is 4.4 compared to BapB_AUR_ 4.6, thus, considerably lower. Secondly, *S. xylosus* is not lowering the pH below 5 in medium containing glucose. Hence, even though amyloidogenic potential of BapB_XYL_ was predicted by several amyloid finders in silico, peptide self-assembly into amyloids is impeded, as a pH close to the pI of BapB_XYL_ is not reached during sugar fermentation of the organism. A final pH of 5 and not lower, after glucose fermentation by *S. xylosus*, is in consistence with our observation that *S. xylosus*, contrary to *S. aureus* [9], did not show any growth in medium with pH 4.5.

While glucose has been discussed to indirectly regulate biofilm formation of *S. aureus* [33] and acidosis related to glucose metabolism is essential for biofilm formation in Bap^+^
*S. aureus* strains [9], it is noteworthy that *S. xylosus* is a well biofilm former under neutral to basic conditions (TSB_N_) and that pH decrease during growth is, in contrast to *S. aureus* V329, not necessary to induce biofilm formation. Additionally, the only reduction in biofilm formation between wildtype and mutant strain we observed was when strain TMW 2.1523 was grown under neutral conditions. Besides the already mentioned fact that the pH drop during growth of *S. xylosus* might not be low enough to induce Bap derived peptide aggregation, one could also argue that Bap might not be expressed in sufficient amounts, especially under acidic conditions, to promote formation of multicellular aggregates. This is substantiated by the results obtained from full proteome analysis which revealed a significantly higher amount of Bap_2.1523_ in neutral (TSB_N_) compared to acidic (Lac^+^) medium. Moreover, even though multiple authors reported good visibility of Bap on SDS gels [7,10], *S. xylosus* Bap was poorly visible on the gel, and only a slight band was detectable for strain TMW 2.1523 when grown in TSB_N_. This leads us to hypothesize that Bap_AUR_ is expressed to a higher extent than Bap_XYL_. Whether Bap_XYL_ visibility on SDS gels is impaired by any processing that occurs, as described for Bap_AUR_ under acidic conditions [9], cannot be ruled out and should be kept in mind.

Another difference between the two species to be considered is the structural organization of the proteins. While Bap of *S. epidermidis*, *S. chromogenes*, *S. hyicus* and *S. simulans* share almost 100% protein similarity with Bap_AUR_ [11], Bap_XYL_ and Bap_AUR_ show only around 50 (our study) to 80% [11] protein identity. Thus, different functions cannot be excluded, especially since the correlation between Bap and staphylococcal biofilm formation has mainly been studied extensively in strain *S. aureus* V329 and a function has been solely assigned to the *N*-terminal region of the protein so far. A large part of the protein (C-terminal part) remains bound to the cell surface, with a still unknown purpose for the cell. Lastly, small but predictively important sequence differences need to be considered, for example two short peptide sequences that have been predicted to show high amyloidogenic potential in Bap_AUR_ [9] are not conserved across the two species. Also, Bap_AUR_ Region D is rich in SD repeats and C-terminal SD repeats have previously been described to be an important structural attribute in staphylococcal surface adhesins [34,35]. Bap_XYL_ lacks any of those SD repeat rich regions.

Proteins containing a YSIRK signal peptide and an LPxTG cell anchor motif have been discussed in different contexts in the past and different functions including peptidase and hydrolase activity have been assigned to those proteins [36,37]. Furthermore, Bhp, a Bap homolog found in S. *epidermidis*, has been speculated to mediate biofilm formation at first [11]; however, disruption of *bhp* did not result in any biofilm reduced phenotypes [38]. Thus, we suggest that biofilm formation in *S. xylosus* does not necessarily require Bap, and that its dominant role in biofilm formation is currently rather restricted to strains of selected species such as *S. aureus* and *S. epidermidis* [9,11]. Our findings further suggest that conclusions on the function of Bap drawn from studies conducted with *S. aureus* V329 should be carefully applied to other organisms carrying Bap orthologs when experimental proof is lacking. Furthermore, it appears that other proteins/mechanisms must be involved in multicellular behavior and biofilm formation of *S. xylosus.*

## Figures and Tables

**Figure 1 microorganisms-09-02610-f001:**
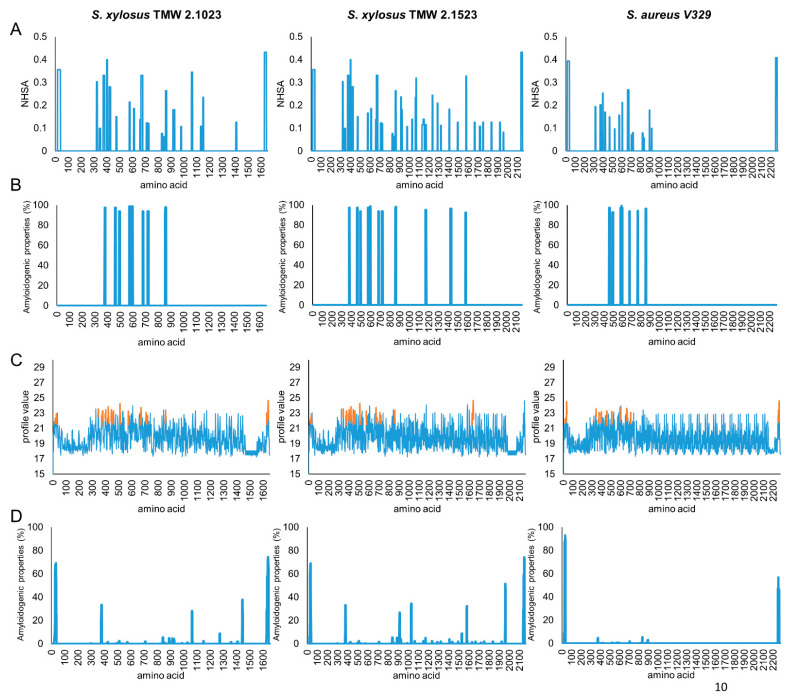
Amyloidogenic structure prediction of different protein sequences from *S. xylosus* TMW 2.1023, TMW 2.1523 and *S. aureus* V329. Prediction based on aggrescan (**A**), waltz (**B**), fold amyloid (**C**) and tango (**D**).

**Figure 2 microorganisms-09-02610-f002:**
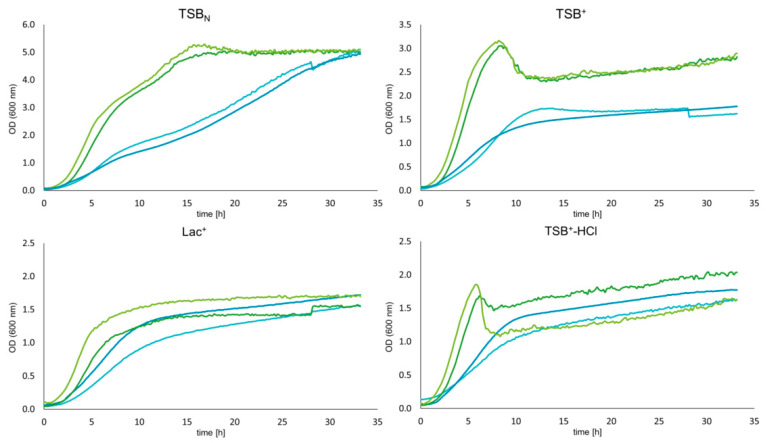
Growth curves of wildtype and mutant strains of *S. xylosus* TMW 2.1023 and TMW 2.1523 in different growth media (as indicated). Curves were recorded in 96-well plates in a plate reader (37 °C, aerobic conditions) every 30 min over a period of 33 h— **–** 023-WT, **–** 023-mut, **–** 523-WT, **–** 523-mut. Data are shown as mean of three biological replicates.

**Figure 3 microorganisms-09-02610-f003:**
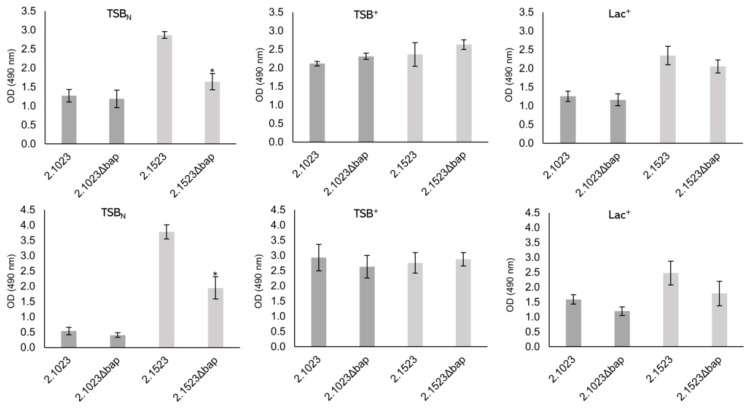
Biofilm formation of *S. xylosus* strains TMW 2.1023 and TMW 2.1523 and its *bap* mutants on hydrophilic (upper row) and hydrophobic (lower row) support in TSB_N_, TSB^+^ and Lac^+^ (as indicated). Significant differences of means are marked by * (*p*-value < 0.05). Mean ± SE.

**Figure 4 microorganisms-09-02610-f004:**
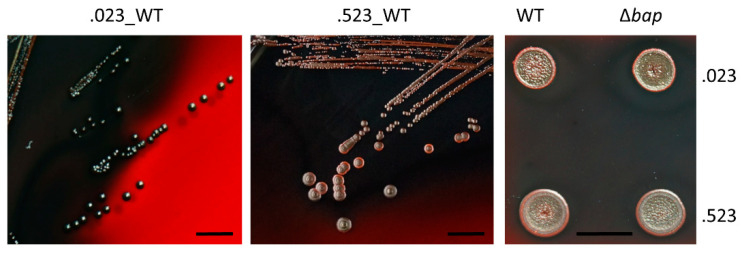
Colony morphology of *S. xylosus* strains on congo red agar. Bacterial overnight cultures were either streaked out for single colonies (**left**, **middle**) or applied as drops to the agar (**right**). Scale bar indicates 10 mm.

**Figure 5 microorganisms-09-02610-f005:**
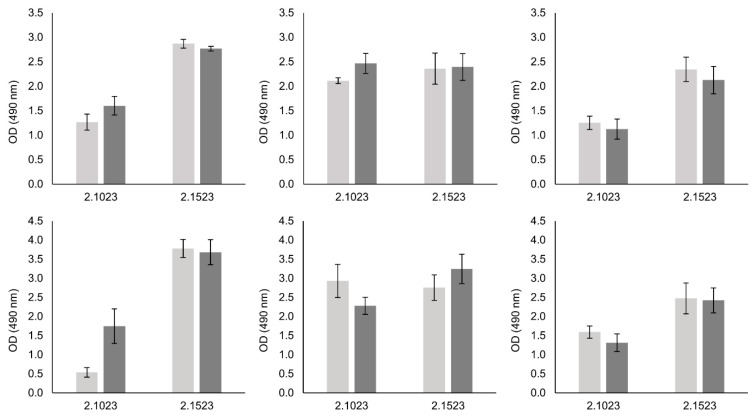
Effect of calcium on biofilm formation of *S. xylosus*. Biofilm formation was quantified of *S. xylosus* TMW 2.1023 and 2.1523 in three different media on hydrophilic (upper row) and hydrophobic (lower row) support. CaCl_2_ (dark grey bars) was added to the respective growth medium to a final concentration of 20 mM.

**Figure 6 microorganisms-09-02610-f006:**
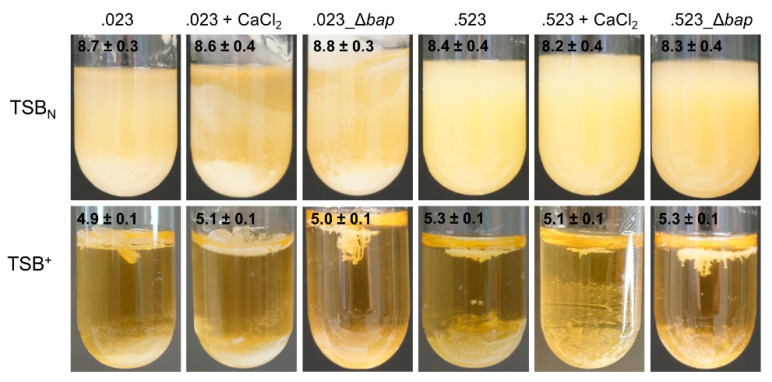
Aggregation behavior of *S. xylosus* TMW 2.1023 (.023), TMW 2.1523 (.523) and the corresponding *bap* mutants grown for 24 h in TSB^+^ at 37 °C (200 rpm). Measured pH values are specified in the top left corner. CaCl_2_ was added to a concentration of 20 mM when indicated.

**Table 1 microorganisms-09-02610-t001:** Sequence comparison of Bap originating from *S. xylosus* TMW 2.1023, TMW 2.1523 and *S. aureus* V329. Different characteristics such as protein size, length of each region of the multidomain protein, molecular weight (MW), isoelectric point (pI) as well as number (^#^) of repeats and EF-hand motifs are listed.

Bap	*S. xylosus* 2.1023	*S. xylosus* 2.1523	*S. aureus* V329
Accession	JGY91_02455	JGY88_01140-45	AAK38834
Length total (aa)	1651	2161	2276
YSIRK Signal Peptide	1–44	1–44	1–44
Region A	316	316	316
Region B	458	458	458
Region C (incl. spacer)	644	1160	1321
Region D (incl LPXTG)	189	183	137
TM helix	1624–1641	2134–2151	2249–2266
MW (kDa)	173.1	224.3	238.5
pI_Bap	4.01	3.90	3.90
pI_BapB	4.41	4.39	4.61
^#^ RepeatsA	2	2	2
sequence	TAEDN	TAEDN	AQDDDNIKEDSNTQEESTNTSSQSSEVPQTKK
^#^ RepeatsC	7	13	14
type of C repeats	ig-like domain type 6	ig-like domain type 6	ig-like domain type 3
^#^ RepeatsD	17	16	7
sequence	13× GTGENP, 1× GKGENP, 1× GGGENP, 1× GIGENP,1× GTGENT	14× GTGENP, 1× GAGENP, 1× GTGENT	2× SDDNSDNGNN 1× SDDNSGNGDN 1× SDDNSDN 1× SGAGDTSD 2× SGAGDNSD
%p.identity_Xyl vs Aur	45.13	58.97	-
%p.identity_B_Xyl vs Aur	80.18	79.96	-
^#^ EF motifs	7	7	4
seq_EF2	DYDKDGLLDRYER	DYDKDGLLDRYER	DYDKDGLLDRYER
seq_EF3	DTDGDGKNDGDEV	DTDGDGKNDGDEV	DTDGDGKNDGDEV
%p.identity EF2_Xyl vs. Aur	100	100	-
%p.identity EF3_Xyl vs. Aur	100	100	-

**Table 2 microorganisms-09-02610-t002:** LFQ intensities determined during full proteome analysis for two different *S. xylosus* strains in two different growth media. Indicated are log_2_ fold change of intensity values, adjusted (Benjamini-Hochberg method) *p*-values and whether the change in expression between the compared conditions is considered as statistically significant (α = 0.05, True vs. False). If the log_2_ fold change was <2, true is written in italics.

Biofilm Associated Protein (Bap)			
TMW 2.1023			JGY91_02455
	plankt. vs. sessile, Lac^+^	log_2_ fold change	0.901
		*p*.val (adj.)	0.158
		significant	FALSE
TMW 2.1523			JGY88_01140-45
	plankt. vs. sessile, Lac^+^	log_2_ fold change	1.15
		*p*.val (adj.)	0.0189
		significant	*TRUE*
	plankt. vs. sessile, TSB_N_	log_2_ fold change	5.11
		*p*.val (adj.)	0.0000231
		significant	TRUE
	Lac^+^ vs. TSB_N_ (plankt.)	log_2_ fold change	−2.97
		*p*.val (adj.)	0.000139
		significant	TRUE
	Lac^+^ vs. TSB_N_ (sessil)	log_2_ fold change	0.981
		*p*.val (adj.)	0.0356
		significant	FALSE

## Data Availability

The proteomics raw data, MaxQuant search results and used protein sequence databases have been deposited with the ProteomeXchange Consortium via the PRIDE partner repository [29] and can be accessed using the data set identifier PXD029728. Sequencing data of TMW 2.1023 and TMW 2.1523 is accessible under JAEMUG000000000 and CP066721-CP066725, respectively.

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
