# Peer review of "Bap-Independent Biofilm Formation in Staphylococcus xylosus"

_microorganisms, 2021, doi:10.3390/microorganisms9122610_

Round 1

Reviewer 1 Report

This manuscript describes an investigation of biofilm associated protein (Bap) in Staphylococcus xylosus. The functions of S. aureus Bap can not easily be extrapolated to S. xylosus Bap. Thus, Bap appears as non-essential for biofilm formation in S. xylosus.

The manuscript is interesting and well written. The objectives of the study are very clear and are well contextualized with the correct bibliography.  Figures and tables are well-structured and detailed to highlight the results.

I only have a few minor comments. 

  • You use abbreviations very often, be sure to define them once in the text. Examples of not defined abbreviations : CRA line 20 in the abstract or line106, EF-hand motifs line 173, FDR lines 181 and 188, TCEP and CAA line 150, AA line 182, LFQ lines 185-186, etc…

  • Line 25, write “ different and multifactorial mechanisms ” .

  • In “Biofilm formation assays” (page 3) and “Bacterial aggregation assay” (page 3), explain why you add CaCl2 under certain conditions.

  • Similarly, explain why you acidify the TSB medium with lactic acid in “Biofilm formation assays” (page 3).

  • Line 80, replace “the to be deleted sequence” by “the sequence to be deleted”.

  • Line 147, “450 rpm”, write in g or specify which material is used for this experiment.

  • Lines 201-202, explain what regions A, B, C, D, correspond to.

  • Line 206, add “(BapAUR)”.

  • Line 220, replace “whilst” by “while”.

  • Line 236, what picture are you talking about?

  • On figure S2 and lines 259-260, the weak band for TMW 2.1523 is not really visible at the expected position for Bap (224 kDa) in the wildtype strain.

  • Figure 3, you write line 296 “Significant differences of means are marked by *”, specify for which p-value.

  • Figure 4, specify the scale on the photos.

  • Line 312, what protein are you talking about?

  • Lines 345, 360, 371, make two sentences starting the second sentence with “However”.

  • Line 366, replace “2.1523 planktonic cells” by “TMW 2.1523 planktonic cells”.

Author Response

Author's Reply (in blue) to the Review Report (Reviewer 1)

This manuscript describes an investigation of biofilm associated protein (Bap) in Staphylococcus xylosus. The functions of S. aureus Bap can not easily be extrapolated to S. xylosus BapThus, Bap appears as non-essential for biofilm formation in S. xylosus.
The manuscript is interesting and well written. The objectives of the study are very clear and are well contextualized with the correct bibliography.  Figures and tables are well-structured and detailed to highlight the results.
I only have a few minor comments. 

Thank you very much for the detailed, thorough review. We have included your remarks into the manuscript. In the very few cases where we couldn’t implement your comments into the manuscript, we provided a short explanation below.

You use abbreviations very often, be sure to define them once in the text. Examples of not defined abbreviations : CRA line 20 in the abstract or line106, EF-hand motifs line 173, FDR lines 181 and 188, TCEP and CAA line 150, AA line 182, LFQ lines 185-186, etc…

  • Thanks for the hint. We added definitions for all abbreviations used.

It seems though that “EF hand motifs” is not an abbreviation but rather the general term used for the calcium binding amino acid motif and there seems to be no detailed definition for the nomenclature. (http://smart.embl-heidelberg.de/smart/do_annotation.pl?DOMAIN=EFh.).

  • Line 25, write “ different and multifactorial mechanisms ” .
  • Done.
  • In “Biofilm formation assays” (page 3) and “Bacterial aggregation assay” (page 3), explain why you add CaCl2 under certain conditions.
  • We added the explanations. (L 120/121, L125/126)
  • Similarly, explain why you acidify the TSB medium with lactic acid in “Biofilm formation assays” (page 3).
  • We added two sentences addressing the reasons for the media chosen (L 116-119)
  • Line 80, replace “the to be deleted sequence” by “the sequence to be deleted”.
  •  
  • Line 147, “450 rpm”, write in g or specify which material is used for this experiment.
  • The 450 rpm refer to a thermoshaker not a centrifuge, we added that in the manuscript (L 151)
  • Lines 201-202, explain what regions A, B, C, D, correspond to.
  • We added a couple of sentences explaining why we chose to align each region separately (L206-209).
  • Line 206, add “(BapAUR)”.
  • Done
  • Line 220, replace “whilst” by “while”.
  • Done
  • Line 236, what picture are you talking about?
  • Sorry, the expression we used was irritating, we tried to change it to a better, clearer wording (L246).
  • On figure S2 and lines 259-260, the weak band for TMW 2.1523 is not really visible at the expected position for Bap (224 kDa) in the wildtype strain.
  • Yes, we are aware that the band for Bap is very weak and especially when being photographed hardly detectable. Visibility was a little bit better when we hold the gel in our hands in the lab. We have made many attempts (different media, different protein extraction protocols, increased protein concentrations to load on the gel etc.) to increase the visibility of the band but nothing really worked. In the end we think it just somehow reflects/confirms one of the messages made in the paper, that BapXYL is poorly to not visible on SDS gels while other groups have reported clear visibility of the protein on SDS gels when working with Bap-positive S. aureus strains.
  • Figure 3, you write line 296 “Significant differences of means are marked by *”, specify for which p-value.
  • We added the p-value at the indicated position (capture Figure 3, L 307).
  • Figure 4, specify the scale on the photos.
  • We added scale bar and specification to Figure 4.
  • Line 312, what protein are you talking about?
  • Bap-based aggregation -> Bap.
  • Lines 345, 360, 371, make two sentences starting the second sentence with “However”.
  • We have changed that in all three cases, accordingly.
  • Line 366, replace “2.1523 planktonic cells” by “TMW 2.1523 planktonic cells”.
  • Done

Reviewer 2 Report

In the paper entitled "Bap-independent biofilm formation in Staphylococcus xylosus" the authors have performed an excellent work. 

The authors have applied different experimental approaches to elucidate the effect of Bap on biofilm formation in two biofilm-forming S. xylosus strains, TMW 2.1023 and TMW 2.1523, in comparison with Bap of S. aureus V329. They have shown that, despite the identical structural arrangement, Bap from S. xylosus differs in key factors to Bap of S. aureus. 

The paper is very well structured and written, the methods are presented clearly and the discussion and conclusion are consistent with the results obtained.

Therefore, in my opinion, this paper can be accepted for publication in Microorganisms.

Author Response

Thank you very much.